# Screening of Developmental Dysplasia of the Hip in Europe: A Systematic Review

**DOI:** 10.3390/children11010097

**Published:** 2024-01-13

**Authors:** Wojciech Krysta, Patryk Dudek, Łukasz Pulik, Paweł Łęgosz

**Affiliations:** 1Student Scientific Association of Reconstructive and Oncology Orthopaedics, Department of Orthopaedics and Traumatology, Medical University of Warsaw, 02-005 Warsaw, Poland; s077779@student.wum.edu.pl (W.K.); s077700@student.wum.edu.pl (P.D.); 2Department of Orthopaedics and Traumatology, Medical University of Warsaw, 02-005 Warsaw, Poland; pawel.legosz@wum.edu.pl

**Keywords:** developmental dysplasia of the hip, humans, screening, ultrasonography, newborn

## Abstract

Background: Developmental dysplasia of the hip (DDH) is a prevalent orthopaedic disorder in children, and screening methods vary across regions due to local health policies. The purpose of this review is to systematise the different ultrasound screening strategies for detecting DDH in newborns in Europe. Methods: Eligible studies from the PubMed, Embase, and Scopus databases, published between 1 January 2018 and 18 March 2023, were included. The inclusion criteria specified a European origin, a focus on newborn human patients, and information on ultrasound for DDH detection. Results: In total, 45 studies were included, covering 18 countries. Among them, six nations (Austria, Bosnia and Herzegovina, Poland, Slovenia, the Czech Republic, and Germany) perform universal ultrasound screening. The timing of the first ultrasound varies, with Austria and the Czech Republic within the 1st week, Bosnia and Herzegovina on the day of birth, Poland between 1 and 12 weeks, and Germany before the 6th week. The Graf method is the most popular ultrasound technique used. Conclusions: There is no consensus on the optimal DDH detection approach in Europe. Varied screening methods stem from epidemiological, cultural, and economic differences among countries.

## 1. Introduction

Developmental dysplasia of the hip (DDH) encompasses a spectrum of abnormal hip development that includes a dislocated hip that is either reducible or irreducible, hip instability with the femoral head partially or fully dislocated from the acetabulum, and isolated abnormalities found on ultrasound without clinical findings that may present later [1]. During embryonic development, the femoral head attains a spherical shape, femur has a short neck and a primitive greater trochanter. As the labrum develops, the femoral head becomes centrally positioned within the acetabulum. Interaction between a properly positioned femoral head and joint cartilage is crucial for the ongoing normal development of the hip joint following birth [1,2]. Its aetiology is multifactorial, and the presence of risk factors such as breech presentation, family history, a female sex, being firstborn, a high birth weight, and oligohydramnios may be associated with DDH [3,4]. It is one of the most common congenital deformities, and estimates of its incidence are highly variable, ranging from 3.6 in the United Kingdom, 3.8 in Scandinavia, and 25.5 in Spain to 35.8 in Eastern Europe (per 1000) [5].

Neonatal screening programmes aim to diagnose DDH early and provide timely treatment to achieve the best functional results. Conservative treatment in the form of many varieties of orthoses, such as a Pavlik harness, Tubinger splint, and Frejka pillow, is a safe and effective method if DDH is diagnosed early [6,7]. If left untreated, it may cause long-term complications in the form of range of motion (ROM) restriction, leg length discrepancy, gait abnormalities, and osteoarthritis, potentially leading to severe disability [8,9].

There are no internationally agreed guidelines or standards for DDH screening [10]. There are two primary approaches to ultrasound screening for developmental dysplasia of the hip: selective and universal.

The selective ultrasound screening approach involves children with abnormal clinical examination and DDH risk factors [11].

The universal approach incorporates performing ultrasound examinations on all newborns within a specific age range [11]. Austria (1991) [12] and Germany (1996) [13] were pioneers in implementing universal ultrasound screening as part of their national surveillance programmes. Italy [11], Slovenia [14], and the Czech Republic [15] also perform universal screening. In the Netherlands [16], Ireland [17], France [18], Norway [19], Sweden [20], and the United Kingdom [21], selective ultrasound screening programmes have been implemented.

The purpose of this review is to systematise the different ultrasound screening strategies for detecting DDH in newborns in European countries. A comparative analysis allows us to identify similarities and differences in screening protocols, which can lead to standardisations of screening approaches.

## 2. Materials and Methods

The criteria for the selection of articles were as follows: The article had to be of European origin, with a focus on newborn patients. Each study had to contain information on the use of ultrasound imaging to detect DDH. Studies published between 1 January 2018 and 18 March 2023 were eligible for inclusion. The review was registered with PROSPERO (the International Prospective Registry of Systematic Reviews) before its start (CRD42023403185). We used the PRISMA statement for the systematic review report [22].

We searched the PubMed, Embase, and Scopus databases with the search being performed between 1 January 2018 and 18 March 2023.

The keywords that we used were ‘developmental dysplasia of the hip’, ‘humans’, ‘screening’, ‘ultrasonography’, and ‘newborn’. All animal studies were excluded, and only studies of European origin were included in our systematic review.

We pooled the results, deleted duplicates (n = 82), and then manually screened the titles and abstracts to assess the relevance of the abstract and the origin of the article. Each study was independently reviewed by 2 reviewers. The results of the search were inserted into Google Sheets (Google, Mountain View, CA, USA) and analysed using a set of criteria that were required to be extracted from the chosen articles. We included crucial publications that were discovered through means other than our search strategy and added them to the bibliography. The search process is depicted in Figure 1.

The main bias of this study is that individual articles may not adequately reflect the screening strategy adopted by each of the European countries.

## 3. Results

Austria:

Austria was a pioneer in introducing universal ultrasound screening for DDH and created national guidelines starting in 1991 [12]. Mutter-Kind-Pass (MKP) contains a schedule of recommended well-child check-ups and vaccinations, as well as information on breastfeeding, nutrition, and other topics related to infant care [23]. MKP in Austria recommends universal screening for developmental dysplasia of the hip for all newborns. The initial ultrasound test and examination for DDH should be performed within the first week of life, and a second test between 6 and 8 weeks of age should be performed by either an orthopaedic surgeon or a paediatrician [1,23,24,25]. The results are documented in the MKP record along with any other medical information related to the mother’s and child’s wellbeing. Screening approach in Austria is presented in Figure 2.

Bosnia and Herzegovina:

Bosnia and Herzegovina does not currently have an established screening programme [26]; however, the authors of the study conducted a universal physical and ultrasound examination of the hip on the day of birth. All were performed by an orthopaedic surgeon using Graf’s ultrasound method.

Czech Republic:

In the Czech Republic, systematic neonatal hip screening has been performed for many decades [15]. The “triple sieve method” towards detecting DDH consists of three consecutive ultrasound examinations and is performed with the first universal ultrasound performed in the first week of life, the second performed between 6 and 9 weeks of age, and the last one performed between 12 and 16 weeks of age. In case of doubt, a radiograph can be added [27,28]. It is useful only from the third to the fourth month of life of the child [29]. Screening approach in the Czech Republic is presented in Figure 3.

Denmark:

In Denmark, selective ultrasound screening is implemented for DDH. Official national guidelines [30] recommend a clinical examination of the hip by a midwife after birth, which is repeated at a 5-week follow-up by a general practitioner. The Ortolani, Barlow, and Galeazzi signs and the ROM of each hip are checked at each visit. Furthermore, asymmetrical skin folds are also assessed, although they are a nonspecific sign of possible DDH.

National guidelines do not specify the screening process in the event of a positive clinical examination or the presence of risk factors [31]. According to a study by Husum et al. [32], the authors followed the guidelines, and if clinical signs were positive or risk factors were present, patients were referred for a specialised ultrasound scan using the Graf technique. Screening approach in Denmark is presented in Figure 4.

France:

The French national guidelines for DDH screening recommend a clinical examination of all newborns and selective use of ultrasound for patients with at least one DDH risk factor, including breech delivery, a family history of DDH, postural orthopaedic deformities, or clinical abnormality of the hip during follow-up. The clinical examination must be performed at each routine examination until walking age, as its results can vary over time. The examination should include the Barlow and Ortolani tests and an inspection for limb length discrepancy and asymmetry of skin folds. Ultrasound must be performed in selected patients at the age of 1 month using the Graf technique. According to the guidelines, radiography has no role in DDH screening up to three months of age [18].

In the study by Printemps et al. [33], all infants were screened for DDH via clinical examination at birth, and a systematic US examination was prescribed for all of them from 4 to 12 weeks of age (adjusted age used for premature infants). US measurements were made using the Couture and Tréguier method by radiologists. Screening approach in France is presented in Figure 5.

Germany:

In Germany, children undergo medical examinations known as U1 to U9, which help detect any abnormalities in children’s development. U1 happens immediately after birth to ensure that pregnancy happens safely. U2 happens between 3 and 10 days after childbirth; children with risk factors (breech presentation, family history, or foot deformities) or positive clinical examinations should have an ultrasound performed immediately. U3 takes place at 4–5 weeks of age; all patients should have an ultrasound performed at that time to ensure that any necessary therapy begins before the sixth week of life. In case of improper findings on ultrasound or clinical examination, a follow-up is recommended in 4 weeks. Imaging is performed according to the Graf classification, and the clinical examination uses the Ortolani and Barlow tests, among many others [34,35,36,37]. Screening approach in Germany is presented in Figure 6.

Greece:

The Greek Paediatric Society adopted the most recent guidelines from the American Academy of Pediatrics for DDH screening in infants. These guidelines recommend selective ultrasound screening at 3–4 weeks of age if a positive physical examination is observed and from 6 weeks to 6 months of age for children with risk factors only. Radiography becomes a viable diagnostic tool between 4 and 6 months of age [38,39].

In the study by Touzopoulos et al. [38], a clinical examination was performed on all patients by a paediatrician shortly after birth, who referred patients with suspicion of DDH. All infants were submitted to ultrasound imaging, which was performed using the Graf and Harcke methods by a radiologist. The average age of the patients at the time of referral was 2.2 months. Screening approach in Greece is presented in Figure 7.

Hungary:

Hungarian newborns undergo clinical examination within the first 72 h of birth and subsequently at 3 weeks and 6–8 weeks of age. An ultrasound screening is selectively performed only for infants who have a positive clinical examination or who are at risk due to factors such as breech presentation, macrosomia, family history, or foot deformities [40].

In the study by Gyurkovits et al. [40], the authors decided to evaluate a universal approach to screening for developmental dysplasia of the hip, given their previous practice of selective ultrasound screening. In the universal strategy, the first ultrasound was typically conducted on the third day by an orthopaedic specialist using the Graf method. Newborns with hips classified as IIc or worse received follow-up ultrasounds at 3 and 6 weeks of age and were monitored until 1 year of age. In addition to ultrasound screening, patients underwent a physical examination that included Barlow and Ortolani tests. Screening approach in Hungary is presented in Figure 8.

Ireland:

In 2016, recommendations for screening for DDH were published [41]. Their Implementation Pack recommends selective ultrasound screening for infants with identified risk factors (first-degree family history or a breech position) or abnormal clinical exams. Infants considered at-risk should undergo an ultrasound by 6 weeks of age, with a referral to an orthopaedic clinic if necessary. Those with positive clinical signs within 72 h of birth should have an ultrasound by 2 weeks of age with a follow-up scan at 6 weeks. Babies who show abnormal clinical findings during the recommended 6-week check should also receive an ultrasound within 2 weeks. After 3–4 months of age, confirmation requires an X-ray [17,41].

In the study by Irvine et al. [17], the programme’s practice differs from the guidelines in that all at-risk babies who have a normal ultrasound are then followed up with an additional radiograph at 6 months.

In the study by Mulrain et al. [42], the authors performed an ultrasound at 6 weeks as per the guidelines. For immature (Graf IIa) hips, imaging was repeated at 3 months of age, and then a radiographic review was performed at 6 months of age only for patients that showed normal hips on ultrasound. Screening approach in Ireland is presented in Figure 9.

Italy:

Currently, in Italy, there are no official national guidelines for screening for DDH. All newborns receive a clinical examination from a neonatologist or paediatrician at birth that should be repeated in the first 6 months of life during health evaluations [29]. Newborns with a ‘clunk sign’ must undergo an ultrasound examination before discharge from the hospital or within the first week of life.

Ultrasound examinations have been shown to be more sensitive in detecting all children with DDH than clinical examinations alone. Therefore, efforts are being made to organise a regional universal DDH screening programme, including all newborns, with a hip ultrasound performed at 4–6 weeks of life [29].

In one study by Buonsenso et al. [43], children underwent ultrasound examinations as soon as possible if the clinical examination was positive or in the sixth week of life if the clinical examination did not reveal any abnormalities.

In another study [44], an ultrasound examination was performed on all newborns at around three months. In both studies, the Graf technique was the preferred screening method. Screening approach in Italy is presented in Figure 10.

The Netherlands:

The Dutch national screening programme has been in place since the 1980s [45]. Newborns undergo clinical screening at one week of age, followed by additional checks at one month and three months of age at the health centre [11]. If clinical instability of the hip is detected, an ultrasound screening should be performed within two weeks after referral. If risk factors are present, such as a family history of DDH, breech presentation, female sex, or twin birth, an ultrasound should be performed at 12 weeks of age. The Graf ultrasound technique is the recommended screening method [16,45]. Screening approach in the Netherlands is presented in Figure 11.

Norway:

In Scandinavia, a selective ultrasound screening strategy is preferred over a universal one [19].

In the study by Håberg et al. [46], the authors applied a selective screening strategy with US examination using the Terjesen method and follow-up in 2–3 weeks in case of uncertain results.

According to Norwegian studies [31], a selective screening strategy consisting of a clinical examination of all children and selective ultrasound scanning based on risk factors is recommended. 

In the study by Olsen et al. [19], a universal screening programme was implemented. Clinical and ultrasound examinations took place in the first three days of life. Patients with immature hips were followed up with a rescan every four weeks. Researchers found that adding universal ultrasound to clinical screening performed by the same experienced paediatrician doubled the treatment rate without affecting the already low number of late cases. Screening approach in Norway is presented in Figure 12.

Poland:

There are no official guidelines or recommendations for DDH screening in Poland. The initial ultrasound is typically conducted within the first 1 to 12 weeks after birth [47].

In the study by Pulik et al. [47], universal ultrasound screening was performed. The authors recommended the first ultrasound at 6 weeks of life or in case of a positive physical examination, which included hip orthopaedic examinations and general examinations (performed at birth) or present risk factors (female sex, caesarean section, breech presentation, family history, and physical signs) in the first weeks of life. The second visit was recommended at 12 weeks of age. Screening approach in Poland is presented in Figure 13.

Slovenia:

In Slovenia, a universal ultrasound screening approach is used. It is recommended to perform a clinical examination in the first few days of life using the Ortolani and Barlow tests and the Galeazzi sign on all infants. If the hip is stable and there are no risk factors, an ultrasound (according to Graf) and clinical examination at 6 weeks of age are recommended. In case of a positive clinical examination or if there are risk factors present (breech presentation, family history, foot deformities, or torticollis), an ultrasound in the maternity ward is recommended, with consultation from an orthopaedic surgeon within 2 weeks for patients with a positive clinical examination. For infants with risk factors only, in case of an abnormal ultrasound, consultation in 3 weeks is recommended [14]. 

In the study by Treiber et al. [48], the screening strategy differed from the guidelines. Ultrasound examinations were conducted for all infants within the first week of life, with a follow-up at 12 weeks. Follow-up assessments for immature hips were conducted at six weeks, and for hips classified as pathological (IIc or D), the follow-up occurred at two weeks. Screening approach in Slovenia is presented in Figure 14.

Spain:

Spanish paediatricians formed PrevInfad with the mission of preventing diseases in childhood and adolescence. According to their guidelines, clinical examinations such as the Ortolani and Barlow tests should be performed in the early neonatal period (from birth to seven days of life) [49]. The assessment of hip abduction and asymmetries, such as the Galeazzi sign, should also be checked at each control in the first year of life. An ultrasound should be performed between 4 and 8 weeks of life (X-ray after 3 months of life) in case of a positive clinical examination or two or more risk factors (female sex, breech position, or family history) [50]. Screening approach in Spain is presented in Figure 15.

Sweden:

In Sweden, all newborns undergo a clinical examination for hip instability using the Ortolani and Barlow tests performed by a paediatrician before discharge from the maternity ward. If there is suspicion of dislocation or instability of the hip, the child is referred to an orthopaedic surgeon. The ultrasound examination may be performed using dynamic (Dahlström) or static (Graf) methods. Further clinical hip examinations are performed by general practitioners at child health care centres at 6 to 8 weeks, 6 months, and 10 to 12 months [20]. Screening approach in Sweden is presented in Figure 16.

Ukraine:

In the study by Zinchenko et al. [51], the authors compared selective ultrasound screening to the universal one. Patients were divided into two groups: those with present risk factors or clinical findings and those without risk factors and negative clinical examinations. An ultrasound was performed according to the Graf classification. Implementing a universal screening system was found to be advantageous, as in the selective screening group, 12% of children remained undiagnosed.

United Kingdom:

The UK implemented the Newborn and Infant Physical Examination (NIPE) programme, designed to identify any physical problems in newborns. It comprises a series of hip joint examinations, including the Ortolani and Barlow manoeuvres and Galeazzi signs, as well as a range of motion check-ups carried out in the first few weeks of life [21].

The NIPE screening programme states that babies who show ‘clicky hips’ during physical examination should not be included in the NIPE standards audit but instead should be managed and referred according to the local arrangement [52]. However, the authors of this study provided evidence suggesting that more clinical evaluations should take place following a referral for ‘clicky hips’.

According to the NIPE guidelines, all newborns receive a hip examination as part of their routine physical examination within 72 h after birth, usually in a primary care setting. For babies born in a hospital, the examination should be completed before being transferred home. A second physical examination should be performed at 6–8 weeks to detect any abnormalities that were not evident at birth.

In the event of a positive screening result, an ultrasound should be performed between 4 and 6 weeks of age and reviewed by an orthopaedic specialist before 6 weeks of age. The NIPE guidelines recommend a second clinical examination at 6–8 weeks of age for all patients, regardless of ultrasound results.

The UK’s approach to DDH screening is a selective ultrasound screening using the Graf [53,54,55,56] and Harcke methods [53,54,55,56,57].

The success of a selective screening programme depends on the expertise of the clinicians performing the initial clinical examinations [55]. Screening approach in the United Kingdom is presented in Figure 17.

## 4. Discussion

This systematic review aimed to systematise the different strategies for detecting DDH in Europe. Due to natural differences between countries in the development of healthcare, the economic status of the population, and scientific progress in Europe, not every European country was included, highlighting a lack of sufficient literature. There was also a significant disparity between the amount and quality of research from each country, making it difficult to sufficiently compare each of the screening approaches.

In the study by Poacher et al. [58], insufficient effectiveness of selective screening was observed, confirming the ongoing need for the development and standardisation of the diagnostic process.

In the study by Shorter et al. [59], the issue of the lack of clear recommendations for practical implementation was highlighted. Attention was drawn to the need for more extensive and precise research on this topic.

The absence of universally accepted diagnostic criteria for DDH increases the risk of misdiagnosis, as there is no gold standard test [60].

Ensuring consistency in DDH diagnosis is crucial for providing suitable treatment and minimising variations in standards of care. Minimising variations in diagnostic approaches should lower the diversity observed in DDH management [61].

Standardisation ensures a consistent and uniform approach across healthcare providers and institutions. This streamlined approach makes the diagnostic process more efficient and facilitates effective communication among healthcare professionals. This collaborative approach ensures the sharing of relevant information for comprehensive patient care. Additionally, standardisation allows healthcare systems to track outcomes and adjust protocols based on new evidence or experiences, enhancing the overall quality of care. Overall process transparency and measurability allows for better benchmarking based on performance indicators. Standardisation brings benefits, but its implementation is not easy due to the diversification of the patient population and dynamically evolving medical knowledge. Providing enough specific details to guide everyone yet maintaining a broad scope to encourage collaboration is a key challenge that needs to be balanced. In healthcare, each patient and their case is individual, and it is crucial to always consider situational variations [62].

As a result of the historically increased incidence of DDH in various regions of Europe (e.g., Central Europe), a firmly established diagnostic tradition has emerged, leading to faster advancements compared to other regions. For example, in the Czech Republic, awareness of this disease is high, enhancing the effectiveness of collaboration between doctors and patients. The high adherence of patients allows for an effectively conducted universal screening approach in comparison to other countries [28]. Differences in diagnostic approaches could be a factor contributing to the wide variation in prevalence estimates observed across different geographical locations [61]. Genetic mutations, specific genes, and chromosomal locations influence variations in susceptibility to DDH. Certain HLA A, B, and D types demonstrate an increase in DDH. Newborn swaddling used in many cultures is a risk factor in the development of DDH [5].

The combination of clinical examinations with the use of USG is present in every country in this paper. The difference lies in timing and whether USG is performed universally or after the consideration of various risk factors.

In the study by Husum et al. [32], challenges associated with the subjectivity of clinical examination were underscored. Orthopaedic surgeons exhibited a superior Positive Predictive Value (PPV) in clinical hip examinations in comparison to general practitioners, midwives, and paediatricians. This underscores the importance of specialised training and expertise, particularly in the context of DDH screening programmes.

In the study by Roovers et al. [63], the authors found that, even though the general ultrasound screening programme detected more cases, it did not manage to lower the number of late cases. This study also highlighted the challenge of implementing ultrasound screening after the neonatal period in many countries due to difficulties in ensuring that all children undergo examination.

In this review, we compiled many studies, allowing us to determine which countries screen universally (Austria [1,23,24,25], Bosnia and Herzegovina [26], Poland [47], Slovenia [14,48], the Czech Republic [27,28,64], and Germany [34,35,36,37]) and selectively (the United Kingdom [21,52,53,54,55,56,57], Italy [29,43,44], Denmark [30,31,32], Spain [49,50], the Netherlands [11,45], Hungary [40], Norway [19,46], France [18,33], Ireland [17,41,42], Greece [38], Sweden [20], and Ukraine [51]). Summarised information on DDH screening approaches can be found in Table 1. We also noted the timing of each screening as well as the overall number of them, individual risk factors, and the technique with which the ultrasound was performed.

The number of studies that specifically research the screening part of the treatment of DDH is low. Most of the literature describes incidence and treatment processes and also partly discusses the screening from which we took our data, as well as official government sites that present the guidelines for screening.

A total of 21 out of 43 identified studies favoured the use of universal USG screening.

The potential limitations of this review include the fact that we included papers starting from 1 January 2018, which means that we might have missed some insightful information that could have been released before this date. Additionally, it is worth noting that individual articles may not comprehensively represent the screening strategies implemented by each of the European countries.

## 5. Conclusions

While synthesising the results, it became clear that the literature focused on DDH incidence and treatment processes, with limited dedicated research on screening methodologies. The scarcity of high-quality studies and the absence of national guidelines in some cases highlighted a knowledge gap, emphasising the need for further research to establish a consensus and standardisation. Individual countries (e.g., Austria [23], Denmark [30], Germany [35], the United Kingdom [21], Slovenia [14], and the Netherlands [16]) are implementing national guidelines regarding the diagnosis of DDH.

The lack of uniformity observed across Europe underscores the necessity for standardised DDH screening protocols. This review emphasises the importance of future studies in addressing this variation, promoting collaboration between countries and the development of comprehensive guidelines. Achieving a consensus on optimal screening methods will contribute to early detection, timely intervention, and improved outcomes for infants at risk of DDH.

Future research should aim to bridge the existing gaps in the literature, focusing specifically on the screening aspect. Rigorous studies, encompassing a wider range of countries and ensuring representation from each, will contribute to a more complete understanding of DDH screening practices. Additionally, efforts should be directed towards the development of comprehensive, evidence-based guidelines to guide healthcare professionals across diverse European regions.

## Figures and Tables

**Figure 1 children-11-00097-f001:**
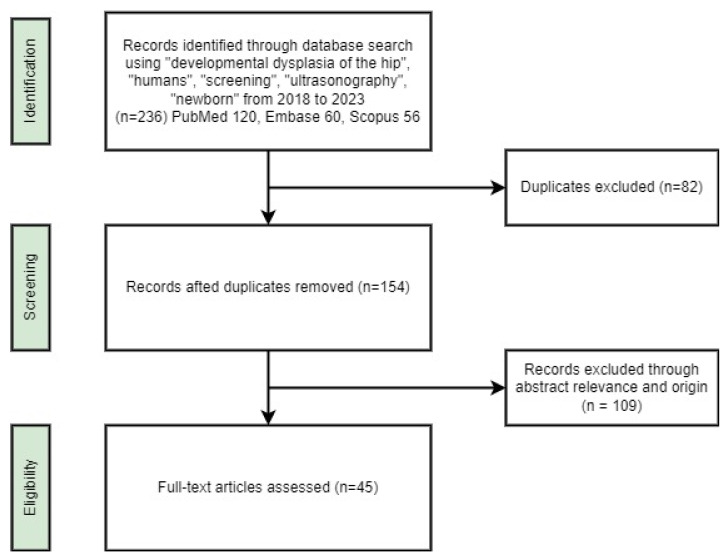
Flow chart of the search process.

**Figure 2 children-11-00097-f002:**
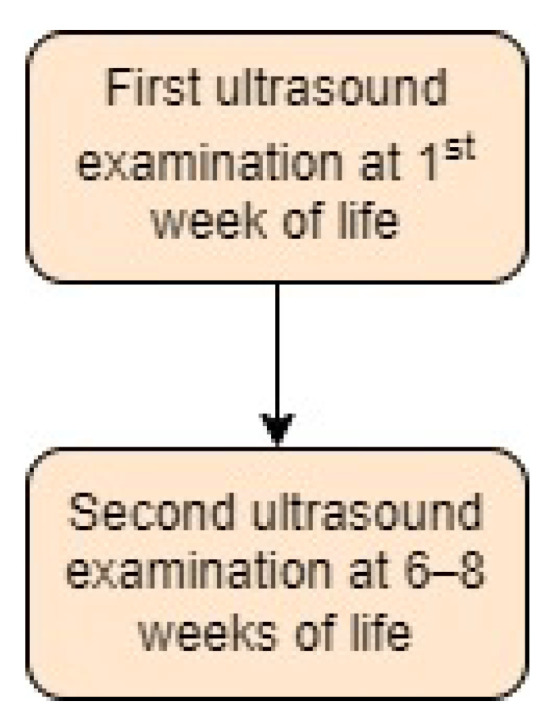
Screening approach for developmental dysplasia of the hip in Austria [23].

**Figure 3 children-11-00097-f003:**
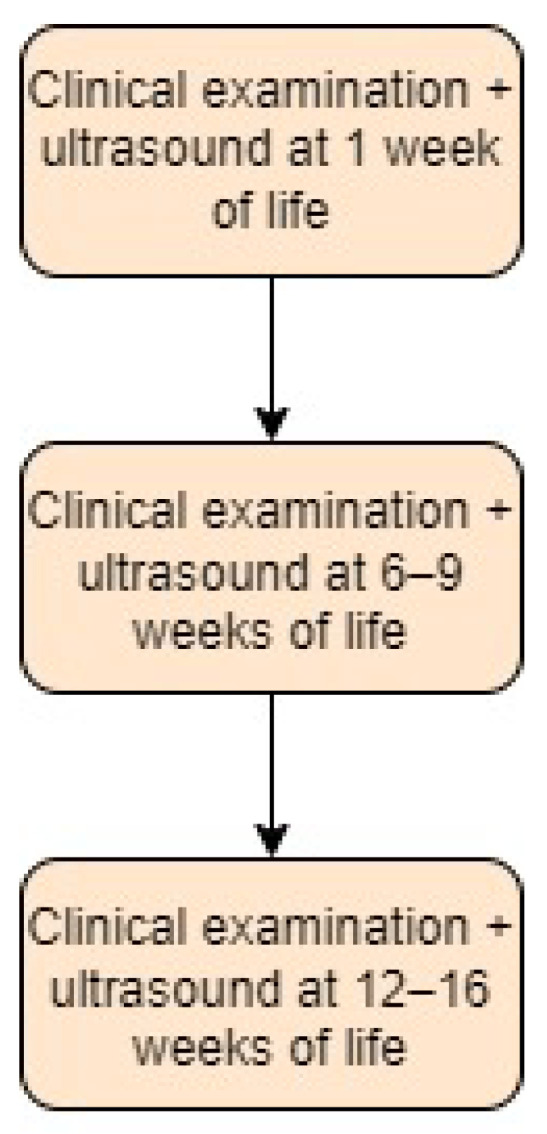
Screening approach for developmental dysplasia of the hip in the Czech Republic [27,28].

**Figure 4 children-11-00097-f004:**
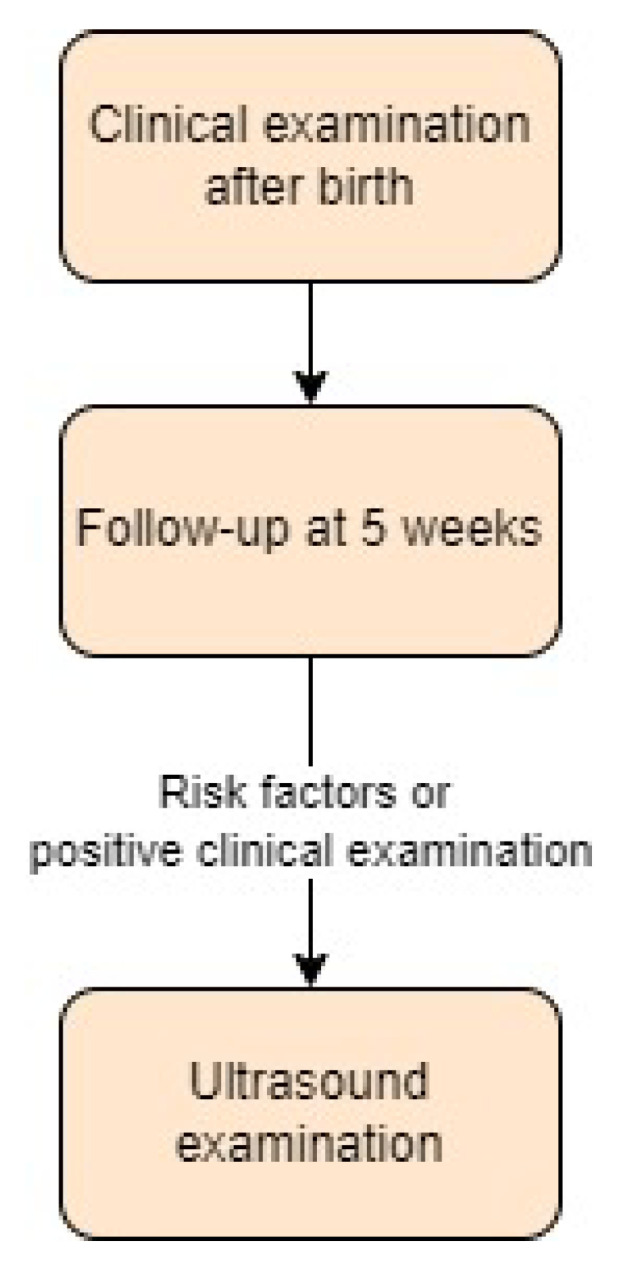
Screening approach for developmental dysplasia of the hip in Denmark [30,31].

**Figure 5 children-11-00097-f005:**
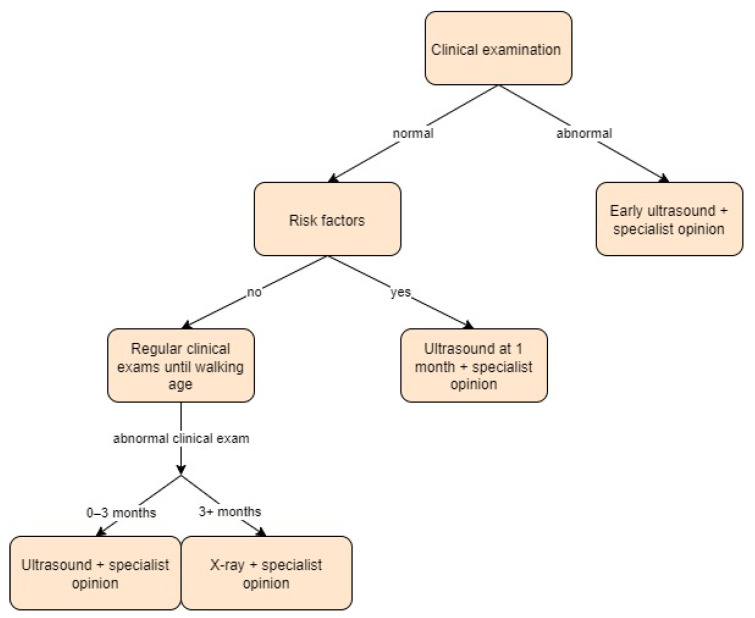
Screening approach for developmental dysplasia of the hip in France [18].

**Figure 6 children-11-00097-f006:**
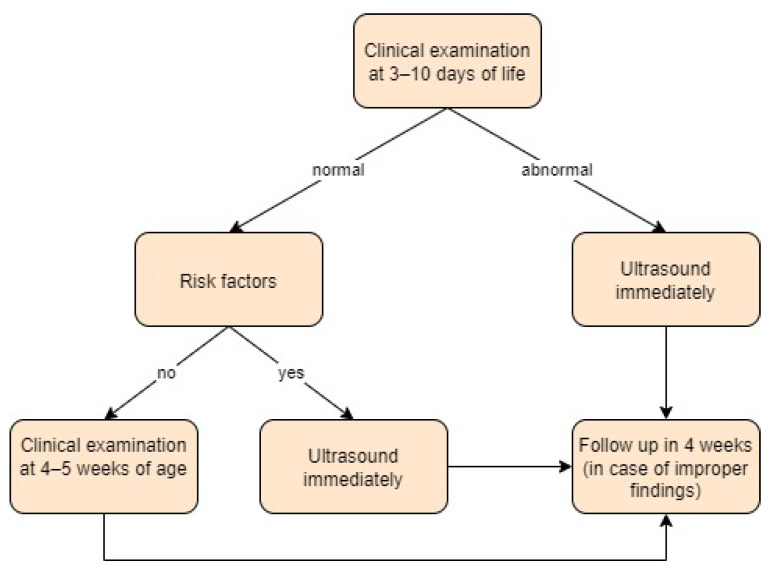
Screening approach for developmental dysplasia of the hip in Germany [34,35].

**Figure 7 children-11-00097-f007:**
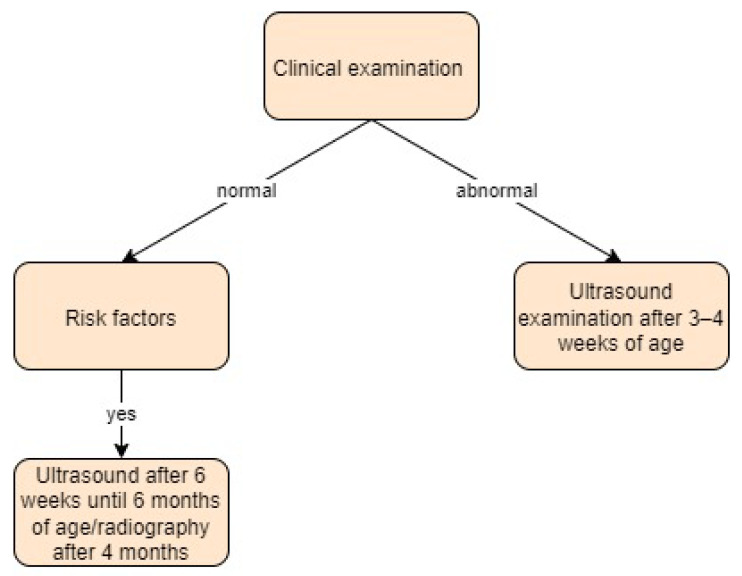
Screening approach for developmental dysplasia of the hip in Greece [38,39].

**Figure 8 children-11-00097-f008:**
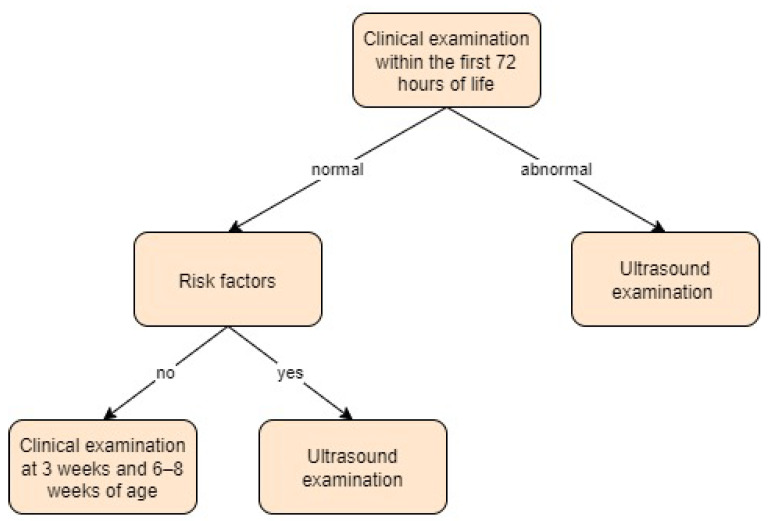
Screening approach for developmental dysplasia of the hip in Hungary [40].

**Figure 9 children-11-00097-f009:**
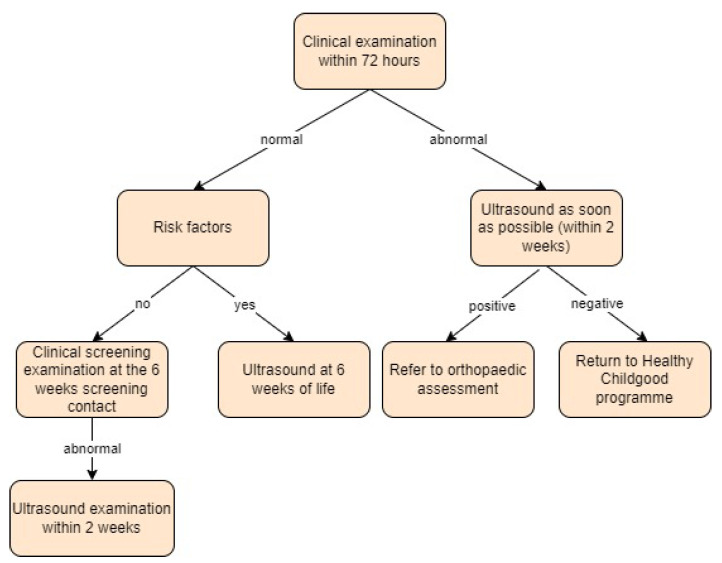
Screening approach for developmental dysplasia of the hip in Ireland [41].

**Figure 10 children-11-00097-f010:**
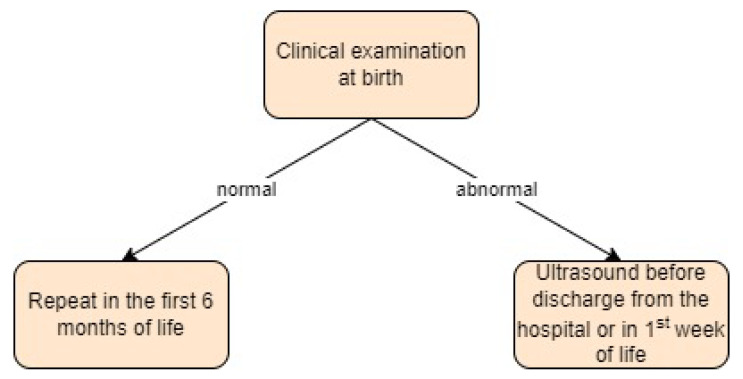
Screening approach for developmental dysplasia of the hip in Italy [29].

**Figure 11 children-11-00097-f011:**
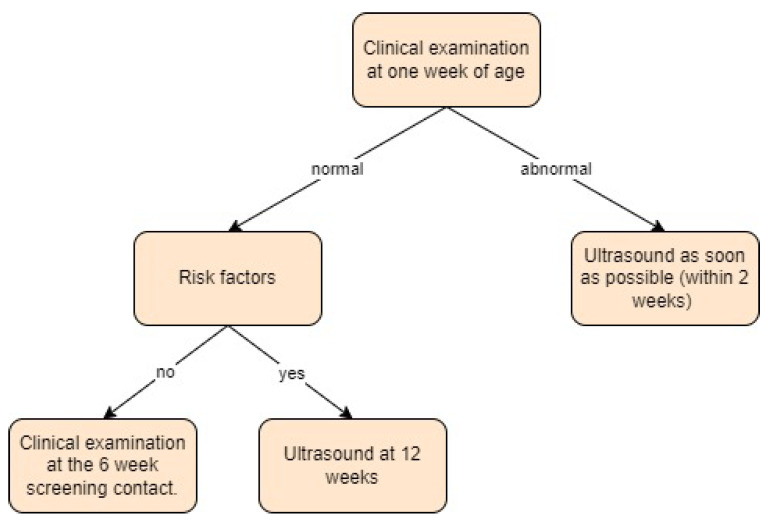
Screening approach for developmental dysplasia of the hip in the Netherlands [11,16].

**Figure 12 children-11-00097-f012:**
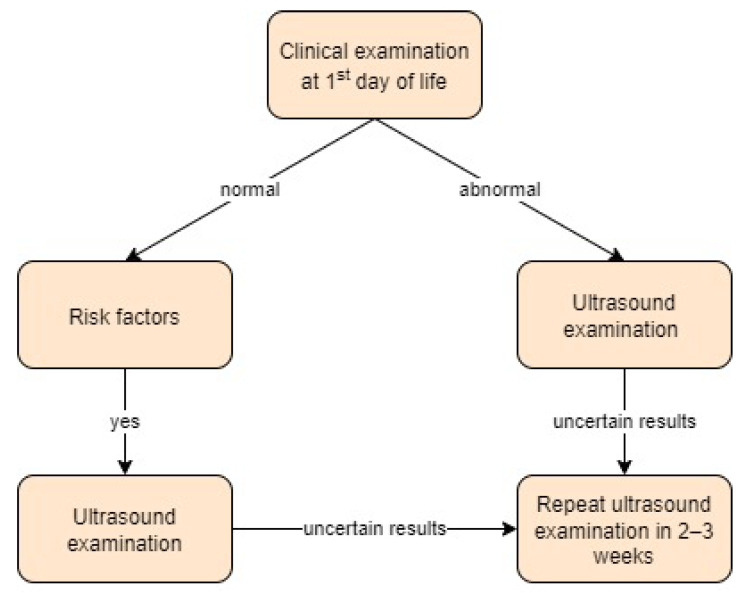
Screening approach for developmental dysplasia of the hip in Norway [46].

**Figure 13 children-11-00097-f013:**
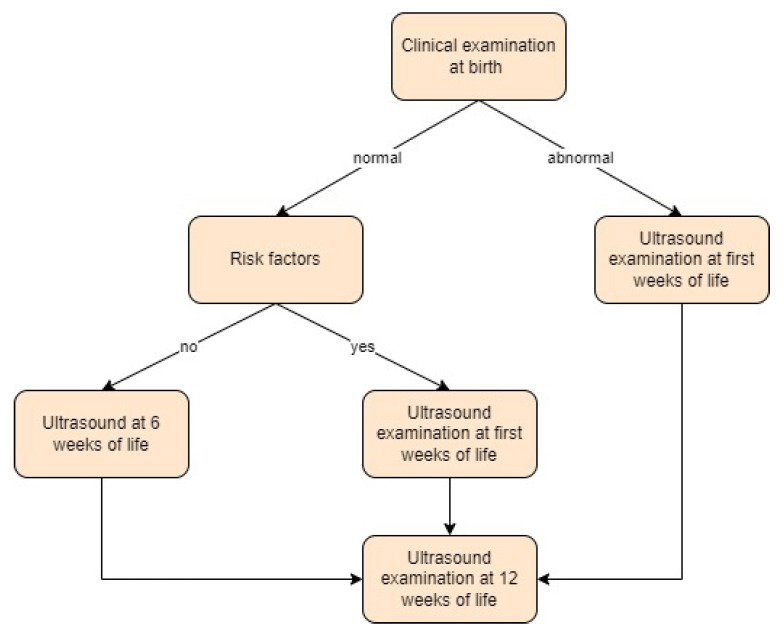
Screening approach for developmental dysplasia of the hip in Poland [47].

**Figure 14 children-11-00097-f014:**
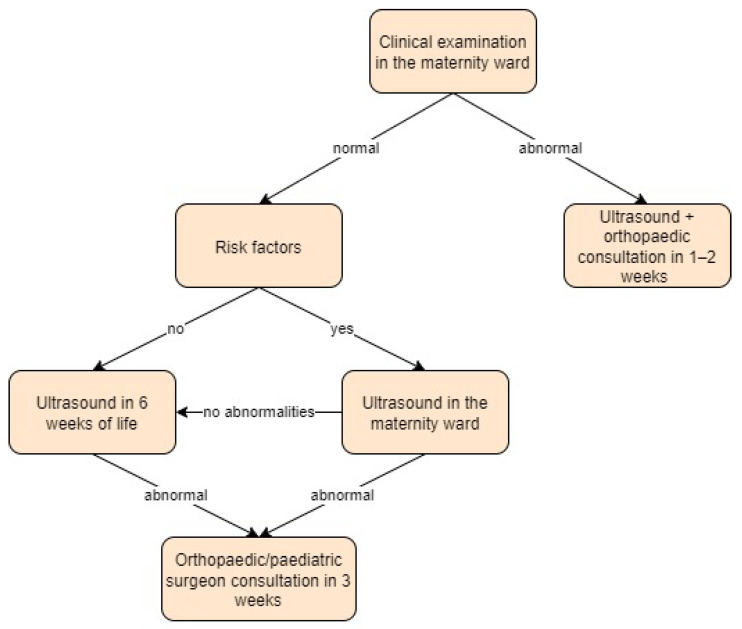
Screening approach for developmental dysplasia of the hip in Slovenia [14].

**Figure 15 children-11-00097-f015:**
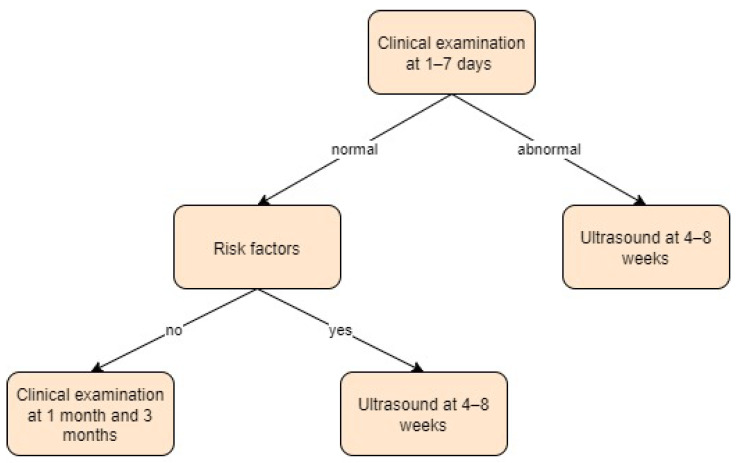
Screening approach for developmental dysplasia of the hip in Spain [49].

**Figure 16 children-11-00097-f016:**
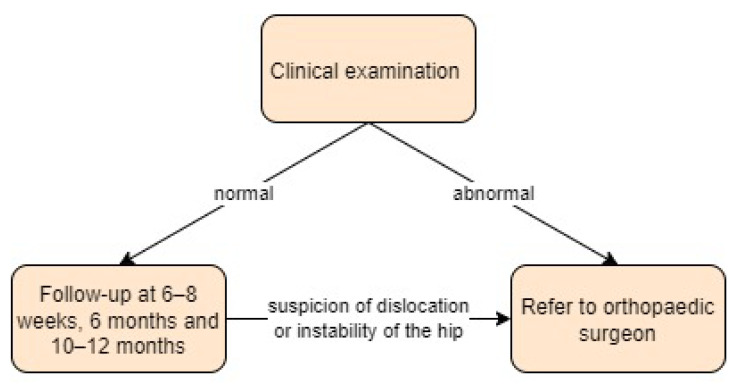
Screening approach for developmental dysplasia of the hip in Sweden [20].

**Figure 17 children-11-00097-f017:**
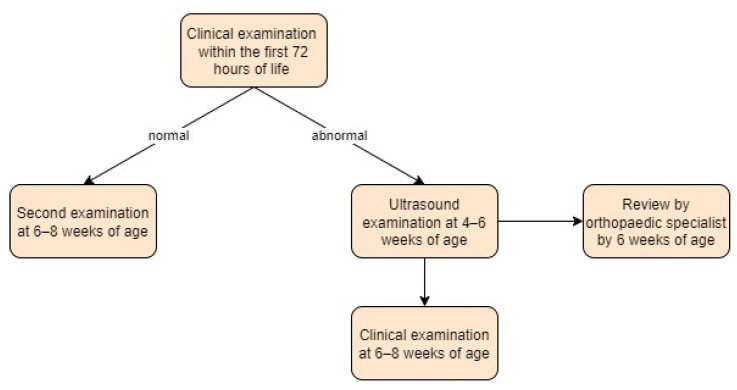
Screening approach for developmental dysplasia of the hip in the United Kingdom [21].

**Table 1 children-11-00097-t001:** Table summarising and comparing DDH screening approaches in different European countries.

Study	Type of Ultrasound Screening	Time of Clinical Examination	Time of Ultrasound Examination	Ultrasound Technique
Austria				
[23]	Universal	1st: 1st week of life 2nd: 6–8 weeks	1st: 1st week of life 2nd: 6–8 weeks	Graf
[24]	Universal	1st: 1st week of life 2nd: 6–8 weeks	1st: 1st week of life 2nd: 6–8 weeks	Graf, manual fixation
[25]	Universal	1st: 2 days after birth 2nd: 6–8 weeks	1st: 2 days after birth 2nd: 6–8 weeks	Graf
[1]	Universal	-	1st: 1st week of life 2nd: 4–7 weeks	Graf
Bosnia and Herzegovina				
[26]	Universal	1st day of life	1st day of life	Graf
Czech Republic				
[27]	Universal	1st: 1st week 2nd: 6–9 weeks 3rd: 12–16 weeks	1st: 1st week 2nd: 6–9 weeks 3rd: 12–16 weeks	Graf
[28]	Universal	1st: 1 week 2nd: 6 weeks 3rd: 4 months	2–3 times	Graf
[64]	Universal	No timeframes	No timeframes	Graf
Denmark				
[30,31]	Selective	1st: 1st day of life 2nd: 5 weeks	-	-
[32]	Selective	1st: 1st day of life 2nd: 5 weeks routinely/40.7 days if +examination	13.7 days	Graf
France				
[18]	Selective	At each routine examination until walking age	1st: 1 month	Graf
[33]	Universal	1st: At birth	1st: 4–12 weeks	Couture and Tréguier
Germany				
[34,35]	Universal	1st: 3–10 days 2nd: 4–5 weeks	1st: 3–10 days if +examination/+risk factors 2nd: 4–5 weeks	Graf
[36]	Universal	1st: 1st week of life	1st: 1st week of life if +risk factors	Graf
[37]	Universal	1st: 1st week of life	1st: 1st week of life	Graf
Greece				
[38]	Selective	1st: shortly after birth	No timeframes	Graf and Harcke
Hungary				
[40]	Universal	1st: <72 h 2nd: 3 weeks 3rd: 6–8 weeks	1st: <72 h IIc and worse: 2nd: 3 weeks 3rd: 6 weeks	Graf
Ireland				
[41]	Selective	1st: <72 h 2nd: 6 weeks	1st: by 2 weeks (only if +examination) 2nd: by 6 weeks (if + risk factors/follow-up for +examination group) After 3–4 months -radiograph	Graf
[17]	Selective	-	1st: 6 weeks + additional radiograph at 6 months	Graf
[42]	Selective	-	1st: 6 weeks 2nd: Graf IIa—3 months +additional radiograph at 6 months	Graf
Italy				
[29]	Selective	1st: at birth and until 6 months	1st week of life	Graf
[43]	Universal	1st: 1st day of life 2nd: 3rd day of life	As soon as possible if clinical findings present or at 6 weeks of age	Graf
[44]	Universal	-	About 3 months	Graf
The Netherlands				
[11]	Selective	1st: 1 week 2nd: 1 month 3rd: 3 months	1st: 3 months/earlier if +examination	Graf
[45]	Selective	1st: 2–3 weeks	1st: 3 months/6 weeks if +examination Other group radiograph—5 months of age	Graf
Norway				
[19]	Universal	1st: <3 days	1st: <3days	Graf
[46]	Selective	1st: 1 day	-	Terjesen
Poland				
[47]	Universal	1st: at birth 2nd: 6 weeks 3rd: 12 weeks	1st: first weeks of life if +examination/+risk factors 2nd: 6 weeks 3rd: 12 weeks	Graf
Slovenia				
[14]	Universal	1st: first few days 2nd: 6 weeks	1st: on maternity ward if +examination/+risk factors 2nd: 6 weeks	Graf
[48]	Universal	-	1st: 1st week of life 2nd: 12 weeks (normal hips), 6 weeks (immature hips), 2 weeks (IIc/D)	Graf
Spain				
[49]	Selective	1st: 1–7 days	1st: 4–8 weeks Radiograph—4–6 months	Graf
[50]	Selective	1st: 48–72 h post-birth	4–8 weeks	-
Sweden				
[20]	Selective	1st: before discharge from a maternity ward 2nd: 6–8 weeks 3rd: 6 months 4th: 10–12 months	No timeframes	Graf Dahlström
Ukraine				
[51]	Universal	-	-	Graf
United Kingdom				
[21,52,53,54,55,56,57]	Selective	1st: <72 h 2nd: 6–8 weeks	4–6 weeks	Graf, Harcke

## Data Availability

Data sharing is not applicable to this article.

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
