# Peer review of "Screening of Developmental Dysplasia of the Hip in Europe: A Systematic Review"

_children, 2024, doi:10.3390/children11010097_

Round 1

Reviewer 1 Report

Comments and Suggestions for Authors

I commend the authors for their outstanding research effort presented in the manuscript titled "Screening of Developmental Dysplasia of the Hip in Europe: A Systematic Review." This comprehensive review explores various ultrasound screening strategies for detecting Developmental Dysplasia of the Hip (DDH) in newborns across European countries. The subject matter addressed in this study is captivating, and the manuscript attains a commendable level of clarity and readability. The introduction is meticulously crafted, the results are presented clearly, and the conclusions are well-founded in the findings.

However, several crucial aspects of the manuscript require further elucidation before it can be considered suitable for publication. Specifically, the discussion section is somewhat sparse, lacking references regarding the optimal timing of ultrasound (US) examinations in newborns and infants. Addressing this gap would allow the authors to propose a universal strategy for DDH screening and early treatment, a crucial aspect shown to reduce the incidence of hip dysplasia in young adults.

In conclusion, while this manuscript exhibits promise, addressing the aforementioned points is imperative to heighten its impact on the field and enhance its suitability for publication. I recommend revision to strengthen these aspects.

Author Response

Reviewer 1 Comment 1: I commend the authors for their outstanding research effort presented in the manuscript titled "Screening of Developmental Dysplasia of the Hip in Europe: A Systematic Review." This comprehensive review explores various ultrasound screening strategies for detecting Developmental Dysplasia of the Hip (DDH) in newborns across European countries. The subject matter addressed in this study is captivating, and the manuscript attains a commendable level of clarity and readability. The introduction is meticulously crafted, the results are presented clearly, and the conclusions are well-founded in the findings.

Authors Response 1: We thank you for the time and effort spent in reviewing our manuscript.

Reviewer 1 Comment 2: However, several crucial aspects of the manuscript require further elucidation before it can be considered suitable for publication. Specifically, the discussion section is somewhat sparse, lacking references regarding the optimal timing of ultrasound (US) examinations in newborns and infants. Addressing this gap would allow the authors to propose a universal strategy for DDH screening and early treatment, a crucial aspect shown to reduce the incidence of hip dysplasia in young adults.

Authors Response 2: Thank you for the suggestions. We appreciate the time that you dedicated. We have expanded our discussion paragraph. Regarding the optimal timing of ultrasound examinations, we used the table to make this data more transparent and easier to read.We included the relevant references.Please check the Microsoft Word file attached.

Reviewer 2 Report

Comments and Suggestions for Authors

I have read this paper with great interest.

I was aware of the different practices on ‘hip screening’, but this paper provides a nice overview on how ultrasound is positioned as part of this ‘hip screening’. This should perhaps be better reflected in the title, as e.g. there are also different practices related to clinical screening practices. Although I value the effort, I have suggested to extract more information (if available) in the retained papers, while perhaps additional search strategies are warranted ?

Could you retrieve some guidelines on the skills needed for clinical examination or to perform ultrasound.

Have you considered to search for other evidence, like NICE guidelines or similar, or other databases, or contacted umbrella organisations like the European Association of Paediatrics ?

On Austria, do I understand it correct that the ultrasound is performed by pediatricians or orthopedic surgeons, and if so, both, or is the first done by a radiologist ?

On figure 3, I assume that the radiography is of limited value on first, and second screening, as ossification is needed ?

How different or confirming is this when compared to the recently published Cureus paper (Alhaddad et al, Cureus 2023). Perhaps the recently published (likely after your search) Poacher et al paper (Bone Jt Open 2023) on the impact of introduction of selective screening in the UK.

Is there value in the Cochrane analysis (Shorter et al, 2011).

Author Response

Reviewer 2 Comment 1: I have read this paper with great interest.

Authors Response 1: Thank you for reviewing our manuscript and your valuable comments.

Reviewer 2 Comment 2: I was aware of the different practices on ‘hip screening’, but this paper provides a nice overview on how ultrasound is positioned as part of this ‘hip screening’. This should perhaps be better reflected in the title, as e.g. there are also different practices related to clinical screening practices. Although I value the effort, I have suggested to extract more information (if available) in the retained papers, while perhaps additional search strategies are warranted ?

Authors Response 2: Thank you for your suggestions. We wanted our title to be as accurate as possible and reflect only what we wanted to focus on (screening, DDH, Europe). The search strategies have been registered prospectively, and they cannot be changed at this stage.

Reviewer 2 Comment 3: Could you retrieve some guidelines on the skills needed for clinical examination or to perform ultrasound.

Authors Response 3: We are aware of the issue of subjectivity in the clinical examination and ultrasound imaging in the diagnosis of DDH. This problem has been discussed in various scientific works (e.g. (Husum et al., 2021, Acta Paediatrica Wiley, Positive predictive values in clinical screening for developmental dysplasia of the hip) ).  We included this information in the discussion section. Please check the Microsoft Word file attached lines 396-405.

Reviewer 2 Comment 4: Have you considered to search for other evidence, like NICE guidelines or similar, or other databases, or contacted umbrella organisations like the European Association of Paediatrics ?

Authors Response 4: In the paragraph about the UK, we utilized the official recommendations of the NIPE. In Austria, we used data from the official government website in accordance with the Mutter-Kind-Pass. In Germany, we also gathered data from official guidelines (U-checkups). We conducted a literature search and there are no guidelines from European Association of Paediatrics.

Reviewer 2 Comment 5: On Austria, do I understand it correct that the ultrasound is performed by pediatricians or orthopedic surgeons, and if so, both, or is the first done by a radiologist ?

Authors Response 5: The ultrasound can be either performed by a paediatrician or an orthopaedic surgeon. We clarified this sentence. Please check the Microsoft Word file attached Line 94.

Reviewer 2 Comment 6: On figure 3, I assume that the radiography is of limited value on first, and second screening, as ossification is needed ?

Authors Response 6: The older the patient, the more valuable X-ray becomes; most studies mention around the 4th month of life. (Agostiniani et al., 2020, Italian Journal of Pediatrics, Recommendations for early diagnosis of Developmental Dysplasia of the Hip (DDH): working group intersociety consensus document). We included this information in the Czech Republic paragraph. Please check the Microsoft Word file attached Line 112-113.

Reviewer 2 Comment 7: How different or confirming is this when compared to the recently published Cureus paper (Alhaddad et al, Cureus 2023). Perhaps the recently published (likely after your search) Poacher et al paper (Bone Jt Open 2023) on the impact of introduction of selective screening in the UK.

Is there value in the Cochrane analysis (Shorter et al, 2011).

Authors Response 7: Thank you for these very interesting articles. Unfortunately, as noted, they were released after the completion of our research. In the study by Alhaddad et al., the entire topic of DDH was covered from anatomical introduction to treatment, and the mention of diagnostics, in our opinion, is insufficient to include in our research focused on screening.

Studies by Poacher et al. and Shorter et al. confirm the lack of specific data and guidelines regarding whether selective or universal ultrasound screening is better. We included these references in the discussion section Lines 358-363.

Reviewer 3 Report

Comments and Suggestions for Authors

- the conclusion section is too small for a such a big study.

- in the introduction section, expand on the subject of DDH abnormalities to provide a more understanding of the condition

- authors should provide more specific data or clarify the factors contributing to these variations in incidence rates to understand the regional differences better

- discuss the challenges and potential risks associated with the lack of international guidelines

- authors should discuss the potential benefits and challenges of standardizing screening techniques

Author Response

Reviewer 3 Comment 1:- the conclusion section is too small for a such a big study.

Authors Response 1: Thank you for your valuable suggestion, we’ve expanded the conclusion section. Please check the Microsoft Word file attached lines 424-442.

Reviewer 3 Comment 2: in the introduction section, expand on the subject of DDH abnormalities to provide a more understanding of the condition

Authors Response 2: Thank you for this valuable suggestion. We added additional information about the embryonic root of the cause to enhance understanding of the condition. We have also expanded the description of the disease. Please check the Microsoft Word file attached lines 27-40.

Reviewer 3 Comment 3:

- authors should provide more specific data or clarify the factors contributing to these variations in incidence rates to understand the regional differences better

Authors Response 3: Thank you for your suggestion, this topic is clearly really interesting. We incorporated the suggestions and added a paragraph in the discussion section about geographical differences. Please check the Microsoft Word file attached lines 380-392.

Reviewer 3 Comment 4

- discuss the challenges and potential risks associated with the lack of international guidelines

- authors should discuss the potential benefits and challenges of standardizing screening techniques

Authors Response 4: Thank you for these really insightful ideas. We incorporated your suggestions into our discussion section. Please check the Microsoft Word file attached lines 358-381.

Round 2

Reviewer 1 Report

Comments and Suggestions for Authors

Well done.

Reviewer 2 Report

Comments and Suggestions for Authors

nothing to add, i agree with the revisions as provided